# Negative Sampling From the Ground Up:
# A Redesign for Recommendations

**Yanbang Wang** [1]   **Jon Kleinberg** [1]   **Yanhong Wu** [2]

## Abstract

Negative sampling is an important yet challenging component in self-supervised graph representation learning, particularly for recommendation systems where user-item interactions are modeled as bipartite graphs. Existing methods often rely on heuristics or human-specified principles to design negative sampling distributions. This potentially overlooks the usage of an underlying "true" negative distribution, which we might be able to access as an oracle despite not knowing its exact form. In this work, we shift the focus from manually designing negative sampling distributions to a more principled method that approximates and leverages the underlying true distribution from the ground up. We expand this idea in the analysis of two scenarios: (1) when the observed graph is an unbiased sample from the true distribution, and (2) when the observed graph is biased with partially observable positive edges. The analysis result is the derivation of a sampling strategy as the numerical approximation of a well-established learning objective. Our theoretical findings are also empirically validated, and our new sampling methods achieve state-of-the-art performance on real-world datasets.

## 1. Introduction

Self-supervised graph representation learning has wide applications in modern recommendation systems. To obtain high-quality embeddings for users and items, the typical method is to model past user-item interactions as a bipartite graph. Node embeddings are often learned in such a way that the distance between a user's embedding and an interacted item's embedding indicates the likelihood of an edge between the two objects.

[1]Department of Computer Science, Cornell University, Ithaca, United States [2]Meta.

*Proceedings of the $43^{rd}$ International Conference on Machine Learning*, Seoul, South Korea. PMLR 306, 2026. Copyright 2026 by the author(s).

Negative sampling is an important step in this learning paradigm: as shown in Figure 1(a), for a given anchor node $u$, a node $v^-$ needs to be sampled from a certain distribution $q^-$, so that node pair $(u, v^-)$ can be treated as a "negative edge" in training, together with the observed positive edge $(u, v^+)$, for training the model via binary loss. Negative sampling is important since it generates all negative samples in the training dataset by certain human-specified rules, which can have huge influence on learned representations. This also applies to more general graphs outside of the bipartite user-item graphs in recommendations.

Negative sampling is not only important but also challenging. Existing works have proposed many heuristics, including random negative sampling (RNS) (Rendle et al., 2012), popularity-based negative sampling (PNS) (Mikolov et al., 2013), hard negative sampling (HNS) (Ying et al., 2018), GAN-based negative sampling (GAN-based NS) (Chae et al., 2018), and in-batch negative sampling (in-batch NS) (Wu et al., 2021), *etc*. More recent theoretical studies (Yan et al., 2024; Yang et al., 2020b) have proposed ideal properties of node embeddings that a good negative distribution should encourage, and then design negative distributions accordingly.

These previous works leave open a critical question: should we consider negative sampling to be a procedure that is primarily up to human heuristics or design, or can we, instead, build the procedure from the ground up — by treating it as a rigorous numerical approximation of certain well-established optimization objective? In this paper, we argue to support the latter, and show a path to achieve it.

We start by formulating and highlighting the concept of the true distribution of negative samples $p^-$, whose objective existence is independent of human choices. As a motivating example, consider that we are in an ideal world where we have enough resources and capability to survey the true opinion of each user $u$'s propensity to like each item $v$ in the entire content pool. This true opinion value can be technically formulated as $P(y = 1|e = (u, v))$, where $y$ is the true class label of the user's like. The Bayes rule gives us distribution $p^+ = P(e|y = 1) \propto P(y = 1|e)P(e)$ and distribution $p^- = P(e|y = 0) \propto (1 - P(y = 1|e))P(e)$, where

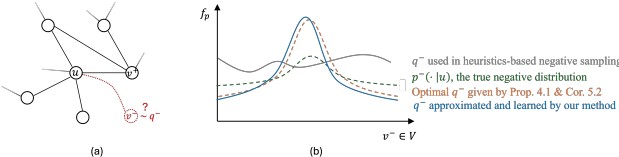

*Figure 1.* **(a)** The negative sampling problem aims to find an optimal distribution $q^-$ to sample negative node $v^-$ from. **(b)** Illustrating the key distributions and idea in our analysis: The green curve $p^-(\cdot|u)$ is the real negative distribution for $v^-$ *w.r.t.* anchor node $u$, which we argue to exist objectively. Our Prop. 4.1 shows that the optimal $q^-$ should be $p^-(\cdot|u)$ reweighed by an exponential factor that relies on the model's output in current epoch. We observe that $q^-$ proposed in previous works, denoted as the gray curve, is mostly heuristics-based and does not leverage the optimal $q^-$, and thus correlates little on distribution landscape. In comparison, our negative sampling seeks to learn and approximate $q^-$, denoted as the blue curve, albeit with inevitable deviation. The bell-shaped curves are only for illustrative purpose and do *not* indicate the distributions to be Gaussian.

$P(e)$ is the node pair prior. In the above formulation, $p^-$ is the true negative distribution that we want to emphasize: every graph from a well-defined domain in practice should have such a $p^-$ that objectively exists. We will provide more rigorous definitions in Section 3.

Of course, the exact form of $p^-$ is unknown in most problems, otherwise the classification problem is already algebraically solvable by reversely deriving $P(y = 1|e)$ from $p^-$, in which case there is no need to do sampling and training. However, $p^-$ being unknown does not necessarily preclude us from being able to access (samples from) $p^-$ as an oracle (though again in many cases there is distribution shift in observation, which we will have a dedicated section to address). The reason that we may still need to generate negative samples from a different distribution than $p^-$ is primarily to reduce variance of learning, *i.e.* to speed up convergence with fewer samples. Some existing analysis (Mukherjee et al., 2011) of classical boosting algorithms such as AdaBoost helps further illustrate this point.

The true negative distribution $p^-$ guides our principled derivation of a good *proposal* negative distribution $q^-$. Illustrated in Figure 1(b), we argue that a good $q$ should be grounded and focused on leveraging the noted true (albeit unknown) negative distribution $p^-$ — including understanding how $p^-$ factors into the expression of a well-established optimization goal, as well as how it can be approximated or recovered if distorted. In fact, an interesting observation is that most existing negative sampling methods make this assumption implicitly without fully exploiting its usage. See Appendix B for more discussion.

**Our work.** In this work, we analyze how to build towards an optimal sampling strategy, by shifting the focus from proposing distributions into crafting sampling procedures based on the argument and the leverage that a true $p^-$ exists, which further expands into two cases. In the first case, we assume that the observed graph is an unbiased sample from the true distribution. $q^-$ can then be derived by drawing its connection to a well-established objective that involves $p^-$. It turns out that the resulted sampling strategy can be interpreted as a form of adaptive hard negative sampling.

In the second case, we assume that the observed graph is a biased sample from the true distribution, and that the positive edges may be only *partially* observable. This case is underexplored in graph learning's literature but has wide applications. For example, in recommendations a positive edge represents a user's manifested interest in an item logged by the platform; however, the absence of an edge can either mean the user truly dislikes the item, or that the user just has not been offered the chance to interact with the item but would have liked it if so. In fact, due to the sparse nature of many complex systems only a small fraction of positive edges may be observable.

To address the second case, the idea is to first derive an unbiased empirical risk estimator depending on *true* distributions, and then convert it into a form that depends only on *observed* distributions, up to some calibration. The calibration, which was deemed a bottleneck in more general settings, can be facilitated by graph topology in a learnable manner. We further validated this new sampling method on real-world datasets and achieved state-of-the-art results.

**Scope.** We primarily consider negative sampling in the context of recommendation and link prediction tasks, following most existing literature on this topic. See Appendix C for more discussion.

**Contributions.** This works makes the following main contributions:

- We propose to fundamentally shift the mode of thinking from proposing negative distributions to principally approximating and utilizing the true distributions in sampling methods for graph representation learning in both recommendations and beyond.

- We theoretically derive and analyze the forms that the optimal sampling strategy should follow, under the assumption that the positive edges are sampled unbiasedly from the true distribution, or that the positive edges are only partially observable.

- We further validate our theorems and new sampling method on real-world data and observe their significant improvements over previous methods.

## 2. Related Work

**Self-supervised Learning on Graphs.** Self-supervised Learning on Graphs aims to train a graph encoder using both the original graph and often some stochastically augmented data, with many applications including recommendations (Shuai et al., 2022), drug discovery (Wang et al., 2021), anomaly detection (Wang et al., 2024), *etc.* Fancy data augmentations are less common for industry-level recommendations due to scalability and efficiency concerns (Yu et al., 2022). In practice, positive samples are usually directly sampled from the edge set or via random walks (Ying et al., 2018), and negative samples by fixing an anchor node and then drawing the other from a certain negative distribution (Yang et al., 2020a).

**Negative Sampling.** Negative sampling was originally proposed as a component in noise contrastive estimation (Gutmann & Hyvärinen, 2010; 2012), and was soon applied to language modeling (Mikolov et al., 2013; Mnih & Teh, 2012) for approximating softmax. It emerged in graph's context when node embedding methods (Perozzi et al., 2014) took inspirations from language modeling. Since then, it has been widely applied in self-supervised graph learning, and in recommendations (Yang et al., 2020a), and soon became a standalone research topic due to its importance.

Existing works on negative sampling have proposed many heuristics, including random negative sampling (RNS) (Rendle et al., 2012; Bordes et al., 2013), popularity-based negative sampling (PNS) (Mikolov et al., 2013; Perozzi et al., 2014), hard negative sampling (HNS) (Huang et al., 2021; Lai et al., 2024), GAN-based negative sampling (GAN-based NS) (Chae et al., 2018; Wang et al., 2018), and in-batch negative sampling (in-batch NS) (Wu et al., 2021; Chen et al., 2020; Zhao et al., 2021), *etc*. More recent theoretical works (Yan et al., 2024; Yang et al., 2020b; Robinson et al., 2021) proposed properties that a good negative distribution should satisfy, such as monotonicity and accuracy. (Shi et al., 2023) analyzes the relationship between hard negatives and BPR loss. (Petrov & Macdonald, 2023) studies over-confidence in negative sampling, and (Chen et al., 2023a) studies the comparison between negative sampling and non-sampling.

Some graph embedding models propose their own negative sampling procedures, *e.g.*, GraphSAGE (Hamilton et al., 2017) and DeepWalk (Perozzi et al., 2014) both use PNS; PinSAGE (Ying et al., 2018) uses a mixture of in-batch NS and HNS customized for large-scale training. In comparison, some other GNN models did not specify their accompanied negative sampling, such as GCN (Kipf & Welling, 2016), GAT (Veličković et al., 2017), MPNN (Gilmer et al., 2017), SGC (Wu et al., 2019). (Ma et al., 2024) provides a comprehensive survey on negative sampling in recommendations.

## 3. Problem Formulation and Definitions

### 3.1. Problem Formulation

The standard task of self-supervised learning on a graph is defined as the following. We are given a graph $G = (V, E)$ where $V$ is the node set, $E$ is the edge set. Denote by $x_u \in \mathbb{R}^m$ the raw features of node $u \in V$. The goal is to learn a graph encoder $f$ defined as a mapping $\mathbb{R}^m \to \mathbb{R}^n$, so that $f(x_u)$ can best represent node $u$ and be used in downstream tasks. $f$ is usually parameterized by a neural network with parameters $\theta$, learned by minimizing the following objective:

$$J = \sum_{\substack{(u,v)\in E, \\ v_1^-,\dots,v_k^- \sim q^-}} \left[ L^+(u,v) + \frac{\tau}{k} \sum_{i=1}^k L^-(u, v_i^-) \right]$$

(1)

where
$$L^+(u,v) = g^+(f(x_u), f(x_v)),$$
$$L^-(u,v^-) = g^-(f(x_u), f(x_{v^-}))$$
(2)

which traverses every edge $(u,v)$ in $E$, picking $k$ negative nodes from the proposal negative distribution $q^-$ for each edge, and collecting the corresponding positive and negative losses. $g^+$ and $g^-$ are simple functions that measure dissimilarity and similarities between the $f$-encoded node embeddings $f(x_u)$ and $f(x_v)$, respectively. $\tau$ is a weighing constant for analysis purpose, assumed to be 1 by default.

**The negative sampling problem asks:**

(⋆) *What's the best $q^-$ to use in objective $J$ above?*

### 3.2. Probabilistic Setup

First, all probabilities are defined in the sample space $\{(e,y,s)\}$ which has the following meaning: (1) Random variable $e = (u,v) \in V \times V$ is a node pair in the graph. (2) Random variable $y \in \{0,1\}$ is the class indicator variable for ground truth. $y = 1$ means that the associated node pair $e$ is a truly positive edge, and $y = 0$ otherwise. (3) Random variable $s \in \{0,1\}$ is the observation indicator variable. $s = 1$ means the edge is observed, and $s = 0$ otherwise.

We can define several distributions based on the specified sample space. $p^+ = P(e|y = 1)$ and $p^- = P(e|y = 0)$ are the true positive distribution and true negative distribution respectively. $\hat{p}^+ = P(e|s = 1)$ and $\hat{p}^- = P(e|s = 0)$ are the observable positive distribution and the observable negative distribution respectively. $\pi^+ = P(y = 1)$ and $\pi^- = P(y = 0)$ are the class priors for the ground truth. $p = P(e)$ is the node pair prior, assumed to be uniform by default, *i.e.* all node pairs are assigned equal weight of consideration initially. $P(y = 1|e)$ represents the link predictor or recommender that we wish to obtain ultimately.

If the data is unbiasedly sampled then $\hat{p}^+ = p^+$, $\hat{p}^- = p^-$. We hold this assumption until Section 6 when we address

the case of biased observation. Also notice that $G$ can either be viewed as containing a set of edges $E$ drawn from distribution $\hat{p}^+$, or equivalently as containing a set of negative edges, $V \times V \backslash E$, drawn from distribution $\hat{p}^-$.

As a convention of notation: since $p^+ = p^+(e) = p^+(u,v) = P(e|y=1) = P(u,v|y=1)$, $p^+$ is essentially a joint distribution of nodes $u,v$, *i.e.* $p^+ : V \times V \to [0,1]$. Hence we can define its corresponding marginal distribution $p^+(u)$, and conditional distribution $p^+(\cdot|u)$. We further use $(u,v) \sim p^+$ to denote that $u,v$ are drawn from the joint distribution, and use $u \sim p^+$ to denote that $u$ is drawn from the marginal distribution. Similar conventions apply to $p^-$, $\hat{p}^+$, and $\hat{p}^-$.

The following proposition gives the generalized objective that negative sampling problem optimizes.

**Proposition 3.1.** *$J$ is the empirical estimation of the following expected risk term*

$$R_1(f) = \mathbb{E}_{u \sim p^+}\big[\mathbb{E}_{v \sim p^+(\cdot|u)}[L^+(u,v)] + \mathbb{E}_{v^- \sim q^-}[L^-(u,v^-)]\big]$$

(3)

### 3.3. Difference between $q^-$, $p^-(\cdot|u)$, and $\hat{p}^-(\cdot|u)$

It is crucial to distinguish the three negative distributions noted above: $q^-, p^-(\cdot|u), \hat{p}^-(\cdot|u)$, which are all defined over node set $V$. $q^-$ is the proposal negative distribution whose optimal form we seek to find, which may or may not depend on the anchor node $u$ though. That is why in the notation we suppress the dependence of $q^-$ on any other entities for the time being. $p^-(\cdot|u)$ is the true negative distribution given anchor node $u$ fixed, which we cannot directly observe if the observations are biased. $\hat{p}^-(\cdot|u)$ is the observed negative distribution, the distribution that our data (*i.e.* the observed negative edges) are directly sampled from.

Our ultimate goal in this paper is to construct $q^-$ from $\hat{p}^-(\cdot|u)$, as introduced in Section 1. Also notice that although the best form of $q^-$ depends on $p$, it does not have to be exactly the same as $p$.

**We provide a notation table in Appendix A.**

## 4. Unbiased Observations

**Roadmap.** This section shows how the best $q^-$ can be principally derived in two steps. First, we introduce another objective $R_2(f)$, in addition to the $R_1(f)$ defined above. By explaining $R_2(f)$, we will show why it is a well-established objective to optimize for training. Second, once we establish the importance of $R_2(f)$, we show there is only one specific $q^-$ that renders $R_1(f)$ equivalent to $R_2(f)$, thereby proving the optimality of our design for $q^-$.

**What is $R_2(f)$?**
Given a graph $G = (V, E)$ with distributions $p^+$ and $p^-$ as defined in Section 3, consider the following risk term $R_2(f)$ that we wish to minimize, associated with encoder $f$ that

has parameters $\theta$:

$$R_2(f) = \mathbb{E}_{(u,v) \sim p^+}\big[-\log\ p_f(v|u)\big] \tag{4}$$

$$p_f(v|u) = \frac{e^{g(f(x_u),f(x_v))}}{e^{g(f(x_u),f(x_v))} + \tau \mathbb{E}_{v^- \sim p^-(\cdot|u)} e^{g(f(x_u),f(x_{v^-}))}} \tag{5}$$

where $p_f(v|u)$ is the generalized softmax probability of node $v$ given anchor node $u$, based on $f$'s output. $g$ can be any similarity measurement such *e.g.* dot product, and at this moment it is to be distinguished from the $g^+$ and $g^-$ in problem formulation. See more discussion in Appendix D.2 for ensuring $p_f$ to be a proper distribution.

The definition of $p_f(v|u)$ above has shown up as a popular objective in many previous works (Robinson et al., 2021; Rusak et al., 2024; Oord et al., 2018; Tian et al., 2020; Chen et al., 2020). For example, if we set $\tau = 1$, then it is the expectation of the classic noise-contrastive estimation (NCE) loss in (Robinson et al., 2021)'s Eq.(1); if we set $\tau = |V| - 1$, it becomes the expectation of the InfoNCE loss - see (Rusak et al., 2024)'s Eq.(1) or the original paper (Oord et al., 2018)'s Eq.(4). .

Also, crucially note that the negative distribution in the denominator of Eq.5 is $p^-$, instead of $q^-$. This is because we desire the probability of each edge to be as large as possible — calibrated against the probabilities of *all* other possible negative edges sampled from the *true* negative distribution $p^-$, rather than against edges from an arbitrary proposal distribution $q^-$. Using $p^-$ for calibration reflects the competition that positive edges face from the entire set (distribution) of negative edges, which represents the general objective we seek to optimize.

**Why is $R_2(f)$ an important objective to optimize?**
$R_2(f)$ can be further interpreted by writing down its empirical estimator:

$$\hat{R}_2(f) = \frac{1}{n} \sum_{\{(u_i,v_i)\}_{i=1}^n \sim \hat{p}^+} \big[-\log\ p_f(v|u)\big] \tag{6}$$

$$= \frac{1}{|E|} \sum_{(u,v) \in E} \big[-\log\ p_f(v|u)\big] \tag{7}$$

which has been widely used in contrastive learning literature as explained above. Besides, minimizing $\hat{R}_2(f)$ is equivalent to maximizing the probability of the following random graph model, assuming uniform node prior $P(u)$. To see this, note that

$$P(G) = \prod_{u,v \in V} P(u,v)^{p^+(u,v)} = \prod_{u,v \in V} [p_f(v|u)P(u)]^{p^+(u,v)} \tag{8}$$

$$= Z \prod_{u,v \in V} [p_f(v|u)]^{\mathbb{1}_{\{(u,v) \in E\}}} \tag{9}$$

where $Z = \prod_{u,v \in V} P(u)^{p^+(u,v)}$ is a constant. Taking log of Eq. 8 gives to $-\hat{R}_2(f)$.

**What $q^-$ can make $R_1(f)$ equivalent to $R_2(f)$?**
If our ultimate goal is to minimize $R_2(f)$, what can we tell about the sampling problem defined in Section 3? The following proposition shows that there exists a unique choice of $q^-$ which ensures that optimizing the $R_1(f)$ in Eq. 4 is identical to optimizing the $R_2(f)$ in Eq. 3.

**Proposition 4.1.** $\nabla_\theta R_1(f) \equiv \nabla_\theta R_2(f)$ *if and only if the following conditions hold:*

- *For each $u \in V$ fixed, $q^-(v) \propto p_f(v|u)p^-(v|u)$*

- $g^+ = -g, g^- = g$

Since $\nabla_\theta R_1(f) \equiv \nabla_\theta R_2(f) \Rightarrow \arg\min_\theta R_1(f) \equiv \arg\min_\theta R_2(f)$, this proposition shows that to optimize $R_2(f)$ the best proposal negative distribution $q^-$ to be used in Eq. 1 should take the simple form of $q^- \propto p_f(\cdot|u)p^-(\cdot|u)$. To approximate $R_1(f)$, $g^-$ should be further reweighed by the partition constant $z_u = \mathbb{E}_{v^- \sim p^-(\cdot|u)}[p_f(v^-|u)]$, which however does *not* need to be explicitly estimated as it cancels off with the $z_u$ for normalizing $q^-$ in sampling. See Appendix D.3 for further discussion.

To analyze $p_f(v|u)$'s implications, first notice that the equivalence above is defined between the two gradients, meaning that the property holds throughout the training of $f$. In other words, $p_f(v|u)$ is a changing term during the training, which is calculated based on the output of the model $f$ in the current training epoch. Also notice that a large $p_f(v|u)$ means that the current model $f$ leans towards classifying $(u, v)$ as positive. Therefore, using $p_f(v|u)$ to reweigh the observed (true) negative distribution $p^-$ is a precise form of *adaptive hard* negative sampling. This theoretical result also intriguingly relates in spirit to some of the previous heuristic-driven sampling methods for adaptive hard negative sampling (Zhang et al., 2013; Robinson et al., 2021; Lai et al., 2024).

**Implementation** According to Proposition 4.1, we need to compute two terms, $p^-$ and $p_f$, in order to draw samples from the proposal distribution $q^-$. To efficiently estimate $p^-$, we resort to the relationship $p = p^+\pi^+ + p^-\pi^- \Leftrightarrow p^- = (p - p^+\pi^+)/\pi^- = (p - \hat{p}^+\pi^+)/\pi^-$. This allows us to exploit the sparsity of observed $\hat{p}$ in practice. Refer to Section 3.2 for more definitions. Here, the positive prior $\pi^+$ is a hyperparameter to tune, and $\pi^- = 1 - \pi^+$. For approximating $\hat{p}^+$, it is typical to use the Monte Carlo method (Robert, 1999), which averages the delta function over all observed neighboring edges: $\hat{P}^+(\cdot|u) \simeq \frac{1}{|\mathcal{N}(u)|} \sum_{v \in \mathcal{N}(u)} \delta_{u,v}(\cdot)$, where $\delta_{u,v}$ is the Dirac delta function, and $\mathcal{N}(u)$ is the neighbors of $u$. To estimate $p_f$, we replace the expectation in its definition of Eq. 5 by sampled average.

# 5. Biased Observations

This section presents a sampling method for the bias case where only *some* positive edges are observable. Section 5.1 motivates the bias setup. Section 5.2 introduces an empirical estimator and its corresponding sampling method. Section 5.3 further describes a graph-learning-based method for estimating a key term in our sampling method.

## 5.1. Fomulating the Bias

Many of the real-world graphs have this interesting property of observation bias: while the observation of an edge often signifies a reliably logged interaction, the absence of an edge could either mean that the edge does not exist in reality, or that the existence of the edge just has not been testified.

For example, in recommendations, the absence of a user-item edge can either mean that the user truly dislikes the item, or that the user just has not been offered the chance to interact with the item – but would have liked it if so. This also applies to other types of interactions such as social interactions and protein interactions, and patient-disease diagnosis relations.

To formulate this bias, recall from Section 3 definitions of the random variables: we consider probabilities in the sample space $\{(e, y, s)\}$ where $e \in V \times V$ is a random node pair, $y \in \{0, 1\}$ is the true class variable, and $s \in \{0, 1\}$ is the observation indicator variable.

It is important to distinguish the meaning between $y$ and $s$. As an example in video recommendations, $e = (u, v)$ represents a user $u$ and a video $v$ that we consider. $y = 1$ means that $u$ *would have* liked $v$ if $u$ is offered the chance to view $v$. Also refer to the "ideal world" explanation in Section 1. $s = 1$ means the fact that $u$ likes $v$ is actually logged by the platform, usually by judging from some predefined interaction metrics by the platform such as the actual action of clicking on the "like" button, long video viewing time, positive comment below the video, *etc*.

The observation bias that we consider stipulates that $s = 1$ only when $y = 1$, or equivalently, $P(s = 0|y = 0) = 1$ by contrapositive. We further define $\phi_{u,v} = P(s = 1|y = 1, e = (u, v))$, which is the probability that edge $e$ gets observed conditioned on itself being a true positive. $\phi_{u,v}$ is also called propensity score in some literature.

## 5.2. Unbiased Estimator From Biased Observations
In this case, directly applying the sampling method over the observable distributions is problematic, because both the $p^+$ in the problem definition (Eq.3), and the $p^-$ in $q^- \propto p_f(\cdot|u)p^-(\cdot|u)$, are now unobservable. If we directly replace them by $\hat{p}^+$ and $\hat{p}^-$, further corrections are needed. In fact, we will show that in this case not only $\hat{p}^-$ needs to be corrected, but so does $\hat{p}^+$. In other words, in order

to approximate $R_2(f)$ under biased observation, we need to build two proposal distributions on top of $\hat{p}^+$ and $\hat{p}^-$: a proposal *positive* distribution $q^+$, and a proposal negative distribution $q^-$. The following theorem elaborates this idea.

**Theorem 5.1.** *Assume $L^+ = -L^-$, then*

$$R_2(f) \equiv \mathbb{E}_{u\sim\hat{p}^+}[\mathbb{E}_{u\sim q^+}[\beta_1 L^+] + \mathbb{E}_{u\sim q^-}[\beta_2 L^-]] \quad (10)$$

*where*

- $c = P(s = 1|y = 1)$, $\phi_{u,v} = P(s = 1|y = 1, e = (u,v))$;

- $q^+(v) \propto [\phi_{u,v} p_f(v|u) + \beta_1]^{-1} \hat{p}^+(v|u)$

- $q^-(v) \propto [\phi_{u,v} p_f(v|u)]^{-1} \hat{p}^-(v|u)$

- $\beta_1 = \frac{\pi^+ c(1-c)}{\pi^-}$, $\beta_2 = \frac{(1-\pi^+ c)c}{\pi^-}$

*(Note that $\hat{p}^+, \hat{p}^-, \pi^+, \pi^-$ have been defined in Section 3.1; $p_f(v)$ has been defined in Eq. 17.)*

Theorem 5.1 directly gives the desired form for the new sampling method that best approximates $R_2(f)$. We would provide pseudocode for the full algorithm at the end of this section. Similar to the unbiased case, here we also see $p_f$ shows up as a reweight to the observed distributions, which carries rich implications – see previous discussion under Proposition 4.1.

**Implementation**. To implement this biased case, we first notice that both $c$ and $\pi^+$ do not depend on $v$, and $\pi^+ c = P(s = 1)$ is a value that can be directly estimated from the data by $\frac{2|E|}{N(N-1)}$. $\pi^+$ remains to be a sampling hyperparameter to tune, and $\pi^- = 1 - \pi^+$. To efficiently approximate $\hat{p}^-$, we resort to the relationship $p = p^+\pi^+ c + p^-(1 - \pi^+ c) \Leftrightarrow \hat{p}^- = (p - \hat{p}^+\pi^+ c)/(1 - \pi^+ c)$, which allows us to exploit the sparsity of $\hat{p}$ in practice. Refer to Section 3.2 for more definitions. The sampling complexity remains the same as that in Sec. 4 — also same as uniform negative sampling.

The only unknown relying on $v$ is the propensity score function $\phi_{u,v}$, discussed in next subsection.

### 5.3. Learning-based Estimation of Propensity Scores Using Graph Features

Here we introduce a learning-based method for approximating the propensity function $\phi_{u,v}$. We will first discuss its interpretation, then explain how it can be approximated by a graph encoder parameterized by a neural network.

**Interpretation.** $\phi_{u,v} = P(s = 1|y = 1, e = (u,v))$ is essentially the observation bias of edge $(u,v)$, also known as *exposure bias* in some literature. Here, our key observation is that the existing literature discussing these biases essentially establishes their strong correlation with various *graph-based features*, such as (1) Popularity bias (Zhang et al., 2021), which may be quantified by node degree $d_u, d_v$,

(2) Position bias (Chen et al., 2023b), which may be quantified by the output of distance function $g$ on the node pair $(u,v)$. In addition, we also conjecture that the general structural information of the $u, v$ may also be correlated factors.

Although these bias terms can be easy to identify, conceptualize, and quantify, their relationship with the propensity score, as encapsulated by $\phi$, is very complex and varied from one domain to another. Therefore, we propose to use a neural network as the universal function approximator for $\phi$.

**Using Graph Features.** Define the approximator for $\phi$ as:

$$\hat{\phi}(u,v) = \sigma(\text{MLP}([d_u; d_v; f(u); f(v); g(u,v)])) \quad (11)$$

where $\sigma$ is a non-linearity function such as sigmoid; the degree features and the embeddings of $u, v$ are concatenated as input to the MLP; $g(u,v) = g^+(f(u), f(v))$, see Sec.3.1. The output $\hat{\phi}(u,v)$ is directly plugged into Eq. 27 so that $\hat{\phi}$ is co-trained with $f$ in an end-to-end fashion. The role of $\hat{\phi}$ in Eq. 27 is very similar to that of the attention mechanism (Veličković et al., 2018): both are a plugged-in module that learns to reweigh samples (tokens) alongside primary training; the goal of both modules is to approximate an underlying, physically motivated function.

**Regularization.** $\hat{\phi}$ can be further regulated to align its behavior with the desired properties of $\phi$. Besides the unit-range output enforced by $\sigma$, we also consider its first moment and consistency:

- First moment: $\mathbb{E}_{u,v\sim p^+}[\phi_{u,v}] = c$ ;

- Consistency: $g(u, v_1) > g(u, v2) \Leftrightarrow \phi(u, v_1) > \phi(u, v_2), \; \forall u, v_1, v_2 \in V$.

The first moment constraint restricts the numerical scale of $\phi$, *i.e.* to be centered around $c$ by expectation - see its proof in Appendix D.5. We softly enforce this by mean-squared loss $(\hat{\phi}(u,v) - c)^2$. The consistency constraint is based on the probabilistic gap theory (He et al., 2018; Gerych et al., 2022), which states the general order-preserving property between the propensity function and the decision function. We implement it by list-wise ranking loss ListMLE (Xia et al., 2008), computed between the two length-$(k + 1)$ lists, $(g(u, v^+), g(u, v_1^-), ..., g(u, v_K^-))$ and $(\hat{\phi}(u, v^+), \hat{\phi}(u, v_1^-), ..., \hat{\phi}(u, v_K^-))$.

**Pseudocode.** The full algorithm is presented in Algorithm 1 in Appendix.

**Complexity**. Despite being an intricate framework, our negative sampling has good time complexity due to importance sampling *i.e.* line 9 in Algorithm 1, which essentially reduces per-step complexity from $O(|V|)$ to $O(k)$. In fact, the only place that our negative sampling introduces time overhead is at learning the propensity function, compared with the simplest uniform negative sampling. However, the

computational cost of the additional MLP usually just a fraction of the main backbone model to learn. Our other debiasing steps involve computation of only some extra constants, as can be both seen from our pseudocode and validated by empirical results in Figure 2.

# 6. Experiment

## 6.1. Experimental Setup

**Dataset & Tasks.** We primarily consider graph-based recommendations framed as link prediction problem. We following common choice of many existing works by using three popular benchmarks whose sizes are among the largest in similar works: MovieLens (Ding et al., 2020), Pinterest (Ding et al., 2020; Geng et al., 2015), and LastFM (Wang et al., 2019). A statistical overview of them can be found in Appendix F.1.

**Baselines.** We compare with 10 baseline methods covering both classical and state-of-the-art methods: DNS (Shi et al., 2023), AHNS (Lai et al., 2024), SENSEI (Yan et al., 2024), MCNS (Yang et al., 2020b), NMNR (Wang et al., 2018), IRGAN (Wang et al., 2017), in-batch negatives with LogQ correction (Chen et al., 2020), MixGCF (Huang et al., 2021), uniformly random sampling (RNS), and popularity-based sampling (PNS) using node degrees.

**Base Models.** The choice of base learning model $f$ is independent from the sampling. We consider two popular models: (1) a node embedding based model (Rendle et al., 2012), essentially performing matrix factorization and preceding many classical embedding methods such as deepwalk (Perozzi et al., 2014), and (2) LightGCN (He et al., 2020), a popular GNN model for recommendation.

**Other Configurations.** More details about data preprocessing and tuning are in Appendix F.2.

## 6.2. Analysis of Performance

Table 1 shows that our method consistently outperforms the baselines. Two factors contribute to the better performance. First, we are the only method whose derivations are provably associated with the ground-truth negative distribution, while no baseline utilizes ground-truth negative distribution. Second, no baseline considers observation bias in sampling, resulting in overestimation of an edge's likelihood to be negative and deterioration of training data quality.

Substantial performance gain is observed on Pinterest. We attribute this to the severity of observation (exposure) Bias in Pinterest compared to other datasets, which our method is specifically designed to rectify. See more discussion in Appendix F.4.

DNS and AHNS represent the strongest baselines. While both baselines probe for hard negatives, our method does *not* adopt that as first principle. It just happens so that the best sampling we derived aligns in form with the hard sampling.

**Ablation Study.** We conduct various ablations to verify the effectiveness of the several key components in our sampling method. The results are reported in Table 2 and analyzed in Appendix F.7. The performance drop in each ablation demonstrates necessity of the designed component.

## 6.3. Negative Sampling for Contrastive learning

Our method is *not* designed for *all* graph tasks. However, for comprehensiveness of study we also evaluate it in contrastive-learning-based node classification, using two classic frameworks as base models: GRACE (Zhu et al., 2020) and GCE (Zhu et al., 2021). We align with the original works on metrics used and rounding of reported numbers. Table 3 shows the result.

We see that our method not only performs well in general, but in some cases it even outperforms the original paper that uses full contrastive loss without any sampling - a phenomenon unobserved on link-based tasks. We conjecture that our debiasing and hard negative selection help the model learn more helpful structural information for node classification - see more discussion in Appendix F.5.

## 6.4. Further Analysis

We further study the complexity and numerical behaviors of all sampling methods on MovieLens dataset.

**Time Complexity.** Fig.2(a) plots the epoch times, placing our method in middle range. Among the baselines, DNS and PNS run relatively slowly due to expansive search and lack of a localized sampling algorithm. We also found that most localized sampling methods run faster, whose sampling only accounts for a small fraction of total time. Fig.2(b) - (d) plot asymptotic times, showing a desirable near-linear trend which justifies Sec.5's discussions on time complexity.

**Model Complexity.** Our method is extremely lightweight in terms of parameter overhead. The only parameters overhead introduced by our method in training is the propensity score estimator (parametrized by an MLP), which is extremely lightweight compared to the recommendation model backbone. We report in Appendix F.6 statistics of the model parameters and memory peak in training, against the simplest baseline RNS. We can see that both the memory and parameter overhead of our method is less than $6\%$. Moreover, our method does not have any inference overhead.

**Convergence.** Fig. 3 (a) plots the training loss curves for all sampling methods. Our method, despite also being a hard negative sampling method with a MLP module to train, converges relatively fast — mainly because it is derived from the ground up to strictly approximate the log-softmax loss $R_2(f)$ in an unbiased manner. In comparison, no heuristics-based method approximates a concrete loss term. Note that the absolute loss values are not directly comparable since their expressions vary.

*Table 1.* Performance on real-world datasets measured by Recall@20 (%). Results on NDCG are reported in Appendix F.3. MovieLens and Pinterest have data splits to simulate biased observation, and LastFM to simulate unbiased observation. $^*$ means the best performance; $^{**}$ means statistical significance; underline means the second best result. Baseline MixGCF is designed for GNN base models only.

| Negative Sampling | Base Model: Node Embedding (Rendle et al., 2012) | | | Base Model: LightGCN (He et al., 2020) | | |
|---|---|---|---|---|---|---|
| | MovieLens | Pinterest | LastFM | MovieLens | Pinterest | LastFM |
| RNS | 7.11 ± 0.16 | 7.50 ± 0.19 | 5.97 ± 0.13 | 8.94 ± 0.11 | 7.98 ± 0.25 | 6.39 ± 0.10 |
| PNS | 8.41 ± 0.10 | 8.42 ± 0.05 | 6.64 ± 0.11 | 7.58 ± 0.05 | 8.67 ± 0.05 | 7.10 ± 0.09 |
| LogQ(Chen et al., 2020) | 9.97 ± 0.10 | 8.14 ± 0.09 | 4.74 ± 0.15 | 11.55 ± 0.20 | 11.53 ± 0.05 | 5.53 ± 0.11 |
| DNS (Shi et al., 2023) | 12.42 ± 0.09 | 8.56 ± 0.06 | 7.37 ± 0.21 | 11.89 ± 0.08 | 10.77 ± 0.13 | 7.26 ± 0.12 |
| SENSEI (Yan et al., 2024) | 10.19 ± 0.06 | 9.01 ± 0.10 | 5.13 ± 0.25 | 8.10 ± 0.11 | 10.78 ± 0.14 | 5.44 ± 0.21 |
| MCNS (Yang et al., 2020a) | 8.53 ± 0.08 | 8.68 ± 0.17 | 6.65 ± 0.20 | 7.22 ± 0.05 | 8.85 ± 0.11 | 6.88 ± 0.12 |
| AHNS (Lai et al., 2024) | 12.08 ± 0.09 | 10.84 ± 0.09 | 7.01 ± 0.10 | 12.73 ± 0.10 | 11.35 ± 0.09 | 6.90 ± 0.13 |
| IRGAN (Wang et al., 2017) | 10.57 ± 0.12 | 9.11 ± 0.09 | 6.86 ± 0.09 | 10.23 ± 0.25 | 9.79 ± 0.26 | 7.13 ± 0.11 |
| NMRN (Wang et al., 2018) | 8.76 ± 0.19 | 7.67 ± 0.11 | 6.87 ± 0.11 | 10.22 ± 0.05 | 7.62 ± 0.06 | 7.12 ± 0.14 |
| MixGCF (Huang et al., 2021) | - | - | - | 12.60 ± 0.07 | 10.25 ± 0.08 | 5.63 ± 0.07 |
| Our Method | 12.77 ± 0.13$^{**}$ | 12.59 ± 0.12$^{**}$ | 7.38 ± 0.11$^*$ | 17.41 ± 0.19$^{**}$ | 17.61 ± 0.20$^{**}$ | 7.39 ± 0.12$^{**}$ |

*Table 2.* Ablations of different components of our method.

| Ablation | MovieLens | Pinterest |
|---|---|---|
| (∗) Full method (control group) | 12.77 ± 0.13 | 12.59 ± 0.12 |
| (A) Removing adaptive reweight | 7.60 ± 0.15 | 6.54 ± 0.20 |
| (B) Constant propensity score | 10.85 ± 0.19 | 9.38 ± 0.18 |
| (C) Propensity w/o regularization | 10.46 ± 0.40 | 8.99 ± 0.51 |
| (D) Uninformative class prior | 11.19 ± 0.11 | 9.39 ± 0.15 |
| (E) Sampling for unbiased case | 10.35 ± 0.18 | 8.01 ± 0.15 |

*Table 3.* Negative sampling applied to graph contrastive learning methods.

| Method | GRACE (Zhu et al., 2020) | | GCE (Zhu et al., 2021) | |
|---|---|---|---|---|
| | Cora | CiteSeer | Wiki-CS | Amazon-Photo |
| Full (control) | 83.3 ± 0.4 | 72.1 ± 0.5 | 78.30 ± 0.00 | 92.49 ± 0.09 |
| RNS | 82.6 ± 1.0 | 70.4 ± 1.0 | 76.82 ± 0.03 | 90.46 ± 0.14 |
| PNS | 82.1 ± 0.8 | 71.3 ± 0.5 | 77.03 ± 0.05 | *not converged* |
| DNS | 81.3 ± 0.3 | 70.8 ± 0.6 | 77.78 ± 0.03$^*$ | 92.91 ± 0.10$^*$ |
| AHNS | 82.9 ± 0.4 | 71.0 ± 0.5 | 77.24 ± 0.07 | 90.98 ± 0.14 |
| Our Method | 83.6± 0.6 | 71.6± 0.8 | 77.41± 0.04 | 91.94± 0.15 |

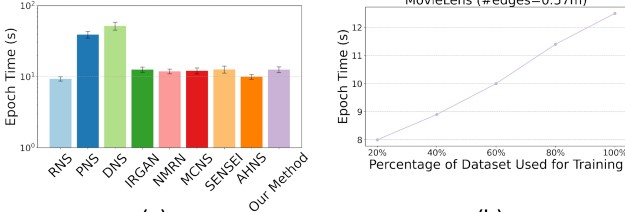
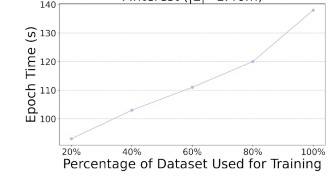
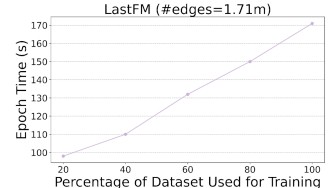

(a)      (b)      (c)      (d)

*Figure 2.* Time complexity of our method: (a) Epoch times compared with other baselines on MovieLens; (b)-(d) Asymptotic time consumption tests on all datasets.

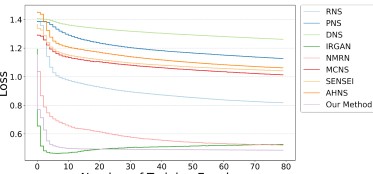
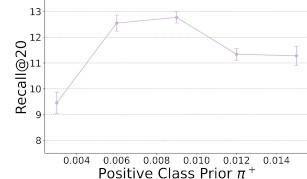
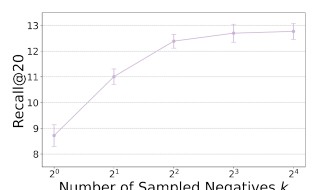

(a)      (b)      (c)

*Figure 3.* Numerical behaviors of sampling methods: (a) convergence curves of training losses; (b) model performance affected by positive class prior $\pi^+$; (c) model performance affected by the negative sampling number $k$. The experiments are done on MovieLens.

**Sensitivity to Hyperparameters.** We investigate two core hyperparameters: positive class prior $\pi^+$, and negative sampling number $k$. The results are shown in Figs.3(b) - (c). Fig.3(b) shows a sweet region for $\pi^+$. Note our theories suggest that we can directly observe (estimate) $P(s = 1)$ from data in an unbiased manner; this value is ~ 0.0027 for MovieLens, according to the plot. Since the positive class prior is $\pi^+ = P(y = 1)$, the ratio $P(s = 1)/\pi^+$ stands for the observation rate which is always below 1 in practice due to various biases. The sweet region of $\pi^+$ indicates that real observation rate for MovieLens might be ~ 20% − 35%, though unfortunately it is impossible to directly verify this.

We further study the effect of negative sampling number $k$

in Figure 3 (c). The result is consistent with our expectation that the performance quickly plateaus as $k$ increases.

# 7. Conclusion

We introduce a new paradigm for negative sampling for graph-based recommendations. The main novelty is the first-principled derivation that leverages graph topology for learning-based debiasing of biased observations. Experiments show advantages over existing approaches, validating our theories and offering directions for better sampling algorithms.

## Impact Statement

This paper presents work whose goal is to advance the field of Machine Learning. There are many potential societal consequences of our work, none which we feel must be specifically highlighted here.

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

# Appendix

## A. Notations

| Notation | Definition |
|----------|------------|
| *Graph and Model* | |
| $G = (V, E)$ | The graph consisting of node set $V$ and observed edge set $E$. |
| $u, v$ | Nodes in the graph, where $u$ typically denotes an anchor node. |
| $x_u$ | Raw feature vector associated with node $u$. |
| $f_\theta(\cdot)$ | Graph encoder function parameterized by $\theta$. |
| $g(\cdot, \cdot)$ | Similarity function (e.g., dot product) between two node embeddings. |
| $L^+, L^-$ | Loss functions for positive and negative node pairs, respectively. |
| $R(f)$ | Expected risk term for the encoder $f$. |
| *Probabilistic Definitions* | |
| $y \in \{0, 1\}$ | Ground-truth class label indicator (1 for true edge, 0 otherwise). |
| $s \in \{0, 1\}$ | Observation indicator variable (1 if observed, 0 otherwise). |
| $p^+, p^-$ | True positive distribution $P(e|y = 1)$, true negative distribution $P(e|y = 0)$. |
| $\hat{p}^+, \hat{p}^-$ | Observed positive distribution $P(e|s = 1)$, observed negative distribution $P(e|s = 0)$. |
| $q^-$ | Proposal negative distribution used for sampling. |
| $p_f(v|u)$ | Generalized softmax probability of $v$ given $u$ based on current model output. |
| *Bias and Propensity* | |
| $\phi_{u,v}$ | Propensity score $P(s = 1|y = 1, e = (u, v))$; probability of observing a true positive edge. |
| $\pi^+$ | Positive class prior probability $P(y = 1)$. |
| $c$ | Observation rate $P(s = 1|y = 1)$, assumed to be the expected value of $\phi$. |
| $\beta_1, \beta_2$ | Re-weighting constants derived for the unbiased risk estimator. |

## B. Implicit Data Assumptions of Existing Methods

Should we consider negative sampling strategies to be primarily a matter of human design, or does a ground-truth distribution of negative samples $p^-$ exist objectively and independently of our choices (and further that its exact form in the real world cannot be easily articulated by "principles" due to nature's complexity)?

For this question, an interesting observation is that many of the existing methods have implicitly taken taken an ambiguous or mixed stance.

To see this, notice that many previous works for *negative* sampling have left *positive* sampling untouched. This essentially respects the existence of a ground-truth positive distribution in nature, from which the observed data (positive edges) get sampled unbiasedly. However, respecting the observed ground-truth positive distribution is equivalent to respecting its corresponding ground-truth negative distribution (by simply taking complement). This is because any graph in nature can be simultaneously viewed as a collection of positive edges sampled from the positive distribution, or a collection of negative edges sampled from the negative distribution. However, in the meantime those existing methods are proposing new negative distributions.

This seeming self-contradiction can only be explained when we consider the existing methods to have respected the existence of $p^-$, but in meantime be proposing a negative distribution different from their respected ground-truth for other good learning properties such as potentially faster empirical convergence. However, this advantage is at the compromise of unbiased estimation, and often does not have theoretical guarantee.

## C. Further Clarification on the Scope of this Work

**Target tasks**. We do *not* propose the negative sampling method as a general strategy for all graph tasks. In fact, none of the existing literature on negative sampling motivates their design as a one-fit-for-all general strategy. It is also unlikely that there exists an optimal sampling strategy for all graph tasks which have different downstream objectives to optimize

for. Like most of the existing works on negative sampling, our problem formulation in Section 3.1 targets the problem of recommendation and link prediction.

**Incremental or Streaming Updates**. Our method naturally supports incremental or streaming updates with its stream-based algorithmic design – as shown in Algorithm 1 (line 5): it iterates on a sequence of edges, rather than an offline static graph as a whole. Therefore, it naturally allows for online learning, as well as random split of new edge streams (and thus A/B testing).

Our theories and methodology are also agnostic to whether the underlying graph is static or evolving. Although we start with a canonical definition of graph in Sec. 3.1, our entire discussion has switched into a probabilistic edge space since Sec. 3.2. Therefore, our method holds as long as incremental updates do not introduce further distribution shift, a topic outside of the scope of this work.

## D. Proofs and Theoretical Discussion

### D.1. Proof of Proposition 3.1

$$J = \sum_{(u,v)\in E,\ v_1^-,\dots,v_k^- \sim q^-(v)} \left[ L^+(u,v) + \frac{\tau}{k}\sum_{i=1}^{k} L^-(u,v_i^-) \right] \tag{12}$$

$$\simeq \mathbb{E}_{(u,v)\sim p^+}\left[ L^+(u,v) + \tau\mathbb{E}_{v^-\sim q^-(v)}L^-(u,v^-) \right] \tag{13}$$

$$= \mathbb{E}_{u\sim p^+}\left[ \mathbb{E}_{v\sim p^+(\cdot|u)}\left[ L^+(u,v) + \mathbb{E}_{v^-\sim q^-(v)}L^-(u,v^-) \right] \right] \tag{14}$$

$$= \mathbb{E}_{u\sim p^+}\left[ \mathbb{E}_{v\sim p^+(\cdot|u)}L^+(u,v) + \mathbb{E}_{v\sim p^+(\cdot|u)}\left[ \mathbb{E}_{v^-\sim q^-(v)}\left[ L^-(u,v^-) \right] \right] \right] \tag{15}$$

$$= \mathbb{E}_{u\sim p^+}\left[ \mathbb{E}_{v\sim p^+(\cdot|u)}L^+(u,v) + \mathbb{E}_{v^-\sim q^-(v)}L^-(u,v^-) \right] \tag{16}$$

### D.2. Generalized Softmax

The full form of the generalized softmax is: for a selected edge $(u,v)$,

$$p_f(v|u) = \frac{e^{g(f(x_u),f(x_v))} + \tau p^-(v|u)e^{g(f(x_u),f(x_v))}}{e^{g(f(x_u),f(x_v))} + \tau\mathbb{E}_{v^-\sim p^-(\cdot|u)}e^{g(f(x_u),f(x_{v^-}))}} \tag{17}$$

$$p_f(v'|u) = \frac{\tau p^-(v'|u)e^{g(f(x_u),f(x_{v'}))}}{e^{g(f(x_u),f(x_{v'}))} + \tau\mathbb{E}_{v^-\sim p^-(\cdot|u)}e^{g(f(x_u),f(x_{v^-}))}},\ \text{for all } v' \neq v \tag{18}$$

This ensures that the probability distribution under the generalized softmax term in Eq. 5 is proper, because the denominator is not symmetric for all $v \in V$, *i.e.* the selected edge $(u,v)$ is treated differently than all other edge $(u,v')$ where $v' \neq v$. In practice since the support of $p^-(\cdot|u)$ usually does not encompass the positive node $v$ in the selected positive edge $(u,v)$, *i.e.* $p^-(v|u) = 0$, Eq. 17 degrades into Eq. 5.

### D.3. Proof of Proposition 4.1

Denote $g(f(x_u),f(x_v))$ by $g_{uv}$

$$\nabla_\theta R_2(f) = \nabla_\theta\,\mathbb{E}_{(u,v)\sim p^+}\left[ -\log\frac{e^{g_{uv}}}{e^{g_{uv}} + \tau\mathbb{E}_{v^-\sim p^-(\cdot|u)}e^{g_{uv^-}}} \right] \tag{19}$$

$$= \mathbb{E}_{(u,v)\sim p^+}\left[ -\nabla_\theta g_{uv} + \nabla_\theta\log(e^{g_{uv}} + \tau\mathbb{E}_{v^-\sim p^-(\cdot|u)}e^{g_{uv^-}}) \right] \tag{20}$$

$$= \mathbb{E}_{(u,v)\sim p^+}\left[ -\nabla_\theta g_{uv} + \frac{e^{g_{uv}}\nabla_\theta g_{uv} + \tau\mathbb{E}_{v^-\sim p^-(\cdot|u)}e^{g_{uv^-}}\nabla_\theta g_{uv^-}}{e^{g_{uv}} + \tau\mathbb{E}_{v^-\sim p^-(\cdot|u)}e^{g_{uv^-}}} \right] \tag{21}$$

$$= \mathbb{E}_{(u,v)\sim p^+}\left[ -\nabla_\theta g_{uv} + \frac{e^{g_{uv}}\nabla_\theta g_{uv}}{e^{g_{uv}} + \tau\mathbb{E}_{v^-\sim p^-(\cdot|u)}e^{g_{uv^-}}} + \frac{\tau\mathbb{E}_{v^-\sim p^-(\cdot|u)}e^{g_{uv^-}}\nabla_\theta g_{uv^-}}{e^{g_{uv}} + \tau\mathbb{E}_{v^-\sim p^-(\cdot|u)}e^{g_{uv^-}}} \right] \tag{22}$$

$$= \mathbb{E}_{(u,v)\sim p^+}\left[ -\nabla_\theta g_{uv} + \mathbb{E}_{v^-\sim p^-(\cdot|u)}\left[ p_f(v^-|u)\nabla_\theta g_{uv^-} \right] \right] \tag{23}$$

$$= \mathbb{E}_{u\sim p^+}\left[ \mathbb{E}_{v\sim p^+(\cdot|u)}\left[ \nabla_\theta(-g_{uv}) + \mathbb{E}_{v^-\sim p^-(\cdot|u)}\left[ p_f(v^-|u)\nabla_\theta g_{uv^-} \right] \right] \right] \tag{24}$$

Notice that the first summation term is exactly positive sampling, and the second summation term is exactly negative sampling. Substitute $g^+ = -g_{uv} = -g(f(x_u), f(x_v))$, $g^- = z_u g(f(x_u), f(x_v))$ into the equation above:

$$\nabla_\theta R_2(f) = \mathbb{E}_{u \sim p^+}\left[\mathbb{E}_{v \sim p^+(\cdot|u)}[\nabla_\theta L^+(u,v)] + \mathbb{E}_{v^- \sim p^-(\cdot|u)}\left[\frac{1}{z_u} p_f(v^-|u) \nabla_\theta L^-(u,v^-)\right]\right] \tag{25}$$

where the partition $z_u = \mathbb{E}_{v \sim p^-(\cdot|u)}[p_f(v|u)]$. Meanwhile, from the definition of $R_1(f)$ in Eq. 4:

$$\nabla_\theta R_1(f) = \mathbb{E}_{u \sim p^+}\left[\mathbb{E}_{v \sim p^+(\cdot|u)}[\nabla_\theta L^+(u,v)] + \mathbb{E}_{v^- \sim q^-}[\nabla_\theta L^-(u,v^-)]\right] \tag{26}$$

Therefore,

$\nabla_\theta R_1(f) \equiv \nabla_\theta R_2(f)$ if and only if $\forall u, v \in V, \ q = \tau p_f(v|u) p^-(v|u)/z_u \propto p_f(v|u) p^-(v|u)$.

### D.4. Proof of Theorem 5.1

Define $p = P(e)$ to be the node pair prior. We first have the following propositions.

**Proposition D.1.** $p^+ = \frac{c}{\phi_{u,v}} \hat{p}^+$.

*Proof.* $\hat{p}^+ = P(e|s=1) = P(e|y=1)P(s=1|y=1,e)/P(s=1|y=1) = p^+ \frac{\phi(e)}{c}$.  $\square$

**Proposition D.2.** $p^- = \frac{1}{\pi^-}(p - \pi^+ p^+)$.

*Proof.* $p = P(e) = P(e|y=1)P(y=1) + P(e|y=0)P(y=0) = \pi^+ p^+ + \pi^- p^-$.  $\square$

**Proposition D.3.** $p = \pi^+ c \hat{p}^+ + (1 - \pi^+ c) \hat{p}^-$.

*Proof.* $P(s=1) = P(s=1, y=1) = P(s=1|y=1)P(y=1) = \pi^+ c$, so $p = P(e) = P(e|s=1)P(s=1) + P(e|s=0)P(s=0) = \pi^+ c \hat{p}^+ + (1 - \pi^+ c) \hat{p}^-$.  $\square$

Based on the above propositions, we have

$$R(f, u) = \mathbb{E}_{v \sim p^+}[L^+] + \mathbb{E}_{v \sim p^-}[p_f(v)L^-] \tag{27}$$

$$= \mathbb{E}_{v \sim p^+}[L^+] + \mathbb{E}_{v \sim (\frac{1}{\pi^-}(p - \pi^+ p^+))}[p_f(v)L^-] \tag{28}$$

$$= \mathbb{E}_{v \sim p^+}\left[L^+ - \frac{\pi^+}{\pi^-}p_f(v)L^-\right] + \mathbb{E}_{v \sim p}\left[\frac{1}{\pi^-}p_f(v)L^-\right] \tag{29}$$

$$= \mathbb{E}_{v \sim \frac{c}{\phi_{u,v}}\hat{p}^+}\left[\left(L^+ - \frac{\pi^+}{\pi^-}p_f(v)L^-\right)\right] + \mathbb{E}_{v \sim (\pi^+ c \hat{p}^+ + (1 - \pi^+ c)\hat{p}^-)}\left[\frac{1}{\pi^-}p_f(v)L^-\right] \tag{30}$$

$$= \mathbb{E}_{v \sim \hat{p}^+}\left[\frac{c}{\phi_{u,v}}L^+\right] - \mathbb{E}_{v \sim \hat{p}^+}\left[\frac{\pi^+ c}{\pi^- \phi_{u,v}}p_f(v)L^-\right] + \mathbb{E}_{v \sim \hat{p}^+}\left[\frac{\pi^+ c^2}{\pi^- \phi_{u,v}}p_f(v)L^-\right] + \mathbb{E}_{v \sim \hat{p}^-}\left[\frac{(1 - \pi^+ c)c}{\pi^- \phi_{u,v}}p_f(v)L^-\right] \tag{31}$$

$$= \mathbb{E}_{v \sim \hat{p}^+}\left[\frac{c}{\phi_{u,v}}L^+\right] + \mathbb{E}_{v \sim \hat{p}^+}\left[\frac{\pi^+ c(1-c)}{\pi^- \phi_{u,v}}p_f(v)(-L^-)\right] + \mathbb{E}_{v \sim \hat{p}^-}\left[\frac{(1 - \pi^+ c)c}{\pi^- \phi_{u,v}}p_f(v)L^-\right] \tag{32}$$

Substitute definitions of $\beta_1, \beta_2, q^+, q^-$ into Eq. 5.1, we will see it exactly equals $R_2(f)$ in Theorem 5.1.

### D.5. Proof of the First-moment Constraint in Section 5.3

$\mathbb{E}_{u,v \sim p^+}[\phi_{u,v}] = \mathbb{E}_{u,v \sim p^+}[P(s=1|y=1, e=(u,v)] = \sum_{u,v \in V}[P(e=(u,v)|y=1)P(s=1|y=1, e=(u,v)] = \sum_{u,v \in V}[P(s=1, e=(u,v)|y=1)] = \sum_e[P(s=1, e=(u,v)|y=1)] = P(s=1|y=1)x = c$

## D.6. Proof of the Consistency Constraint in Section 5.3

The probabilistic gap theory (He et al., 2018; Gerych et al., 2022) essentially states that $\phi(e) = h(P(y = 1|e) - P(y = 0|e))$, $h'(t) > 0$. Therefore, $\phi(e) = h(2g(e) - 1)$ and $\frac{d\phi}{dg}|_{t=e} = 2h'(g(e)) > 0$. Equivalently, $\phi(e_1) > \phi(e_1) \Leftrightarrow g(e_1) > g(e_1)$.

# E. Pseudocode

Pseudocode for the full algorithm can be found in Algorithm 1.

---

**Algorithm 1** Proposed Debiased Negative Sampling

---

**Input:** graph $G = (V, E)$, where observed positive edges $E = \{(u, v)\}$; graph encoder $f$ with parameters $\theta$ to train; number of negative samples $k$; positive class prior $\pi^+$; number of training epochs $T$; similarity function $g$.

1 Compute constants and node prior $\pi^- \leftarrow 1 - \pi^+$, $c \leftarrow \frac{2|E|}{|V|(|V|-1)\pi^+}$, $p(v) \leftarrow \frac{1}{|V|}, \forall v \in V$

Initialize propensity score function $\phi$ as an MLP with parameters $\theta_\phi$

**for** $t = 1, 2, \cdots, T$ **do**

2     Initialize loss $\mathcal{L} = 0$

    **for** *each* $(u, v)$ *pair* **do**

3        Retrieve $u$'s adjacency list $\mathcal{N}(u) \leftarrow \{v'|(u, v') \in E\}$

       Compute positive distribution $\hat{p}^+(v') \leftarrow \frac{1}{|\mathcal{N}(u)|} \sum_{v'' \in \mathcal{N}(u)} \delta_{v''}, \forall v' \in V$

       Compute negative distribution $\hat{p}^-(v') \leftarrow (p(v') - \hat{p}^+(v')\pi^+c)/(1 - \pi^+c), \forall v' \in V$

       Sample negative nodes $v_1^-, ..., v_k^- \sim \hat{p}^-$ via importance sampling

       Compute learnable propensity score $\phi_{u,v}$ by Eq. (11)

       **for** *each* $v^- \in \{v_1^-, ..., v_k^-\}$ **do**

4           Compute generalized softmax probability $p_f(v^-)$ by Eq. (5)

          Compute propensity score $\phi_{u,v^-}$ and its regularization penalty $L_R$ by Eq. (11)

          Compute loss terms $L_v^+ \leftarrow g(f(u), f(v)); L_v^- \leftarrow -g(f(u), f(v)); L_{v^-} \leftarrow -g(f(u), f(v^-))$

          Aggregate reweighed loss terms $\mathcal{L} \leftarrow \mathcal{L} + \left[\frac{c}{\phi_{u,v}} L_v^+ + \frac{\pi^+c(1-c)}{\pi^-\phi_{u,v}} p_f(v)(-L_v^-)\right] + \left[\frac{(1-\pi^+c)c}{\pi^-\phi_{u,v^-}} p_f(v^-)L_{v^-}\right] + L_R$

5     Update parameters $\theta, \theta_\phi$ by descending the gradients $\nabla_{\theta,\theta_\phi} \mathcal{L}$

---

# F. Experiment

## F.1. Dataset

The sources for the datasets used in this paper are:

- MovieLens (Harper & Konstan, 2015):
  https://grouplens.org/datasets/MovieLens/100k/.

- Pinterest (Geng et al., 2015):
  https://github.com/edervishaj/pinterest-recsys-dataset?tab=readme-ov-file

- LastFM: (Wang et al., 2019):
  https://github.com/xiangwang1223/knowledge_graph_attention_network

*Table 4.* Statistics of datasets

| Dataset | Training Edges | Validation Edges | Testing Edges | Users | Items |
|---|---|---|---|---|---|
| MovieLens (Ding et al., 2020) | 563,112 | 6,028 | 6,028 | 6,028 | 3,533 |
| Pinterest (Ding et al., 2020) | 1,355,844 | 55,186 | 55,186 | 55,186 | 9,916 |
| LastFM (Wang et al., 2019) | 1,198,808 | 256,887 | 256,888 | 23,566 | 28,126 |

## F.2. Data Split & Configurations

**Data Split.** Special care is needed when splitting the edge set into training, validation and testing. The most ideal setup should ensure that $E_{\text{train}}, E_{\text{val}} \sim \hat{p}^+$ and $E_{\text{test}} \sim p^+$.

For the unbiased scenario, we preprocess LastFM dataset to create data splits that simulate the case, which further requires $p^+ = \hat{p}^+$. Therefore the data split is done completely at random, *i.e.* $70\%, 15\%, 15\%$ for training, validation, and testing, respectively.

For the biased scenario, we use MovieLens and Pinterest datasets. Since we have no direct access to $p^+$. we split the edges in temporal order by following the commonly used "leave-one-out" strategy as in (He et al., 2017; Rendle et al., 2012): leaving the most recent and the second most recent edge for each node as testing and validation set, respectively. We believe that using temporal splits is one of the most natural ways to introduce realistic bias to the test.

**Configurations.** The configurations for base models are to be fixed. In this paper, we use Adam optimizer, learning rate $1e - 4$, L2 regularization $1e - 5$, hidden embedding size $64$, mini-batch size $1024$, with early stopping. The GNN model used is a two-layered LightGCN. We use lower-bound clipping at $\epsilon = 5e - 2$ and temperature scaling $T = 1.1$ to further regularize $\phi$'s output logits. We set $\tau = k = 8$ for both our method and baselines to ensure fair comparison. We further tune the class prior $\pi_+$ ranged from $1.5$ times to $20$ times of the directly observable $P(S = 1)$; we set $g^-$ as the dot product and $g^+ = -g^-$. For hyperparameters related to the baseline sampling methods, we follow their reported settings for tuning. All experiments are run on an Nvidia V100 GPU.

## F.3. Performance by NDCG

See Table 5.

*Table 5.* Performance of different sampling methods on real-world datasets, measured by NDCG@20 (%). $^*$ means the best performance; $^{**}$ means the best performance with significance; underline means the second best result.

| Negative Sampling | Node Embedding (Rendle et al., 2012) | | | LightGCN (He et al., 2020) | | |
|---|---|---|---|---|---|---|
| | MovieLens | Pinterest | LastFM | MovieLens | Pinterest | LastFM |
| RNS | 2.82 ± 0.10 | 3.01 ± 0.16 | 2.59 ± 0.18 | 4.00 ± 0.12 | 2.97 ± 0.20 | 2.98 ± 0.11 |
| PNS | 3.57 ± 0.11 | 3.35 ± 0.06 | 3.02 ± 0.14 | 3.21 ± 0.04 | 3.70 ± 0.04 | 1.59 ± 0.07 |
| DNS (Shi et al., 2023) | 4.78 ± 0.07 | 4.44 ± 0.07 | 3.58 ± 0.16 | 4.92 ± 0.05 | 4.90 ± 0.09 | 3.33 ± 0.08 |
| SENSEI (Yan et al., 2024) | 4.01 ± 0.06 | 4.20 ± 0.08 | 2.60 ± 0.23 | 3.54 ± 0.10 | 4.88 ± 0.11 | 2.04 ± 0.19 |
| MCNS (Yang et al., 2020a) | 4.08 ± 0.06 | 4.22 ± 0.09 | 3.02 ± 0.12 | 3.17 ± 0.05 | 3.75 ± 0.08 | 2.73 ± 0.08 |
| AHNS (Lai et al., 2024) | 4.35 ± 0.06 | 4.66 ± 0.08 | 3.38 ± 0.10 | 5.90 ± 0.08 | 5.92 ± 0.08 | 3.38 ± 0.10 |
| IRGAN (Wang et al., 2017) | 4.12 ± 0.10 | 3.39 ± 0.10 | 3.27 ± 0.08 | 4.75 ± 0.16 | 4.04 ± 0.18 | 3.02 ± 0.08 |
| NMRN (Wang et al., 2018) | 3.93 ± 0.11 | 2.76 ± 0.12 | 3.22 ± 0.08 | 4.50 ± 0.06 | 3.11 ± 0.05 | 3.14 ± 0.10 |
| **Our Method** | **5.01 ± 0.09**$^{**}$ | **5.06 ± 0.09**$^{**}$ | **3.70 ± 0.12**$^{*}$ | **7.21 ± 0.13**$^{**}$ | **7.28 ± 0.11**$^{**}$ | **3.52 ± 0.10**$^{**}$ |

## F.4. More Discussion on Performance Gain on Pinterest

We attribute the outstanding performance gain on Pinterest to the severity of observation (exposure) Bias in Pinterest compared to other datasets, which our method is specifically designed to correct.

Pinterest's user-item graph is the sparsest among all datasets: its edge density is $2.6^{-3}$ , compared to $3.7 * 10^{-2}$ for MovieLens – a 7x difference. This means that Pinterest potentially contains more missing interactions, and thus suffers from stronger exposure bias. We also found the optimal hyperparameter of for Pinterest to be the smallest among all datasets, meaning that strongest bias correction is applied by our method, which contributes to the outstanding performance.

We further investigate why the magnitude of exposure bias could be so different for recommendation datasets. We identify one contributing factor in the different natures of the different platform – especially their rating collection mechanism: Pinterest is primarily a social media where the users passively wait for recommendation algorithms to feed them next post to consume and (potentially rate). In comparison, MovieLens and LastFM are movie and music platforms respectively, where users more often proactively search for items to consume and rate. In other words, the user's "awareness" of the item set is higher on MovieLens and LastFM than on Pinterest. Since the essence of exposure bias is unawareness of the item pool, this explains why Pinterest naturally has a higher exposure bias in its data."

## F.5. More Discussion on Results for Contrastive Learning

First, we note that the method that uses all possible negatives essentially assumes that the best negative distribution to use is a uniform distribution over all observed negatives. However, according to our paper's analysis, this uniform distribution is most likely not the ground-truth negative distribution (i.e. the motivated in our paper), nor could it be the optimal negative distribution to sample from for training the model.

In comparison, our method is sampling from a carefully learned negative distribution, although it has a very low sampling number (k=8) that is incomparable to the full dataset. Therefore, we conjecture that our win benefits from that our learned negative distribution is closer to the optimal negative distribution, compared with the uniform distribution implicitly assumed by the method using full negatives. This closeness in distribution compensates for the disadvantage of using a lower sampling number. We thank the reviewer for give us an opportunity to elaborate on this interesting phenomenon.

## F.6. Model Complexity

*Table 6.* Number of Model Parameters

| Dataset | RNS (Baseline) | Our Method | Overhead |
|---|---|---|---|
| MovieLens | $6.13 \times 10^5$ | $6.47 \times 10^5$ | 5.2% |
| Pinterest | $4.20 \times 10^6$ | $4.24 \times 10^6$ | 0.9% |
| LastFM | $4.62 \times 10^6$ | $4.66 \times 10^6$ | 0.8% |

*Table 7.* Peak Memory Usage

| Dataset | RNS (Baseline) | Our Method | Overhead |
|---|---|---|---|
| MovieLens | 2.64 GB | 2.61 GB | 1.1% |
| Pinterest | 2.85 GB | 2.79 GB | 2.1% |
| LastFM | 2.85 GB | 2.82 GB | 1.1% |

## F.7. Ablation Studies

To further understand the contribution of each component, we conduct ablation studies on MovieLens and Pinterest datasets. Table 8 summarizes the different ablations and their results, and we have the following analysis.

*Table 8.* Ablations of different components of our method, tested on MovieLens and Pinterest. The metric is Recall@20 (%).

| Ablation | MovieLens | Pinterest |
|---|---|---|
| **(A)** Removing adaptive reweight | 7.60 ± 0.15 | 6.54 ± 0.20 |
| **(B)** Constant propensity score | 10.85 ± 0.19 | 9.38 ± 0.18 |
| **(C)** Propensity w/o regularization | 10.46 ± 0.40 | 8.99 ± 0.51 |
| **(D)** Uninformative class prior | 11.19 ± 0.11 | 9.39 ± 0.15 |
| **(E)** Sampling for unbiased case | 10.35 ± 0.18 | 8.01 ± 0.15 |
| **(∗)** Full method (no ablation) | 12.77 ± 0.13 | 12.59 ± 0.12 |

In ablation (A), we remove the adaptive reweighing component $p_f(v)$ which essentially functions as a filter for hard negatives. This results in most significant performance drop compared to all methods, which demonstrates its importance and supports our theoretical derivation in Proposition 4.1.

Ablations (B) and (C) study the propensity score estimation function $\phi_{u,v}$. In (B), we replace $\phi_{u,v}$ by its expectation over all possible $(u,v)$'s, which is the constant $c$ defined in Theorem 5.1. In (C), we remove its regularization in training, essentially removing the constraints that regulate $\phi_{u,v}$ to be numerically appropriate. Both ablations are shown to affect performance. we also observe (C)'s gap to be larger, which seems to suggest that a numerically well-behaved but inaccurate $\phi_{u,v}$ is more helpful than a numerically ill-behaved but presumably more accurate $\phi_{u,v}$ in debiasing.

Ablation (D) extends the sensitivity analysis in Section 6.4 by setting the hyperparameter of positive class prior $\pi^+$ to be 0.5. Again this ablation leads to performance drop, which helps support our theoretical derivations.

Finally, Ablation (E) examines the case where we over-simplify the data assumption — by applying our base sampling method derived for unbiased observations to address the biased case. The performance drop implies the importance of considering observation bias in negative sampling.

# G. The Use of Large Language Models

Large Language Models (LLMs) are minimally used in preparing this manuscript and should not be considered a substantial contributor. LLMs have only been used to adjust tables and figures layout, and to ensure grammatical correctness.

