# OpenReview forum: "Negative Sampling From the Ground Up: A Redesign for Recommendation"
_ICML.cc/2026/Conference — ICML 2026 regular_

### Official Review · Reviewer_d1rd · 2026-03-06

**Soundness:** 3
**Presentation:** 3
**Significance:** 2
**Originality:** 3
**Overall Recommendation:** 4
**Confidence:** 4

**Summary:**

In contrast to previous methods that rely on heuristics or human-specified principles to design negative sampling distributions, this paper proposes approximating and leveraging the underlying true distribution from the ground up. It addresses two scenarios: unbiased and biased observations. The method's improvements are validated from both theoretical and experimental perspectives.

**Compliance With Llm Reviewing Policy:**

Affirmed.

**Final Justification:**

I also acknowledge the perspectives raised by other reviewers. I maintain my recommendation.

**Key Questions For Authors:**

Please refer to Weakness and Limitation section.

**Limitations:**

(1) The assumption of p- independent of observation is unverifiable.
(2) The intermediate module for estimation p- increase complexity, but gains are marginal.
(3) More experiments and analysis are needed according to Weakness 3.

**Strengths And Weaknesses:**

Strength:
This paper proposes a novel perspective on negative sampling, shifting from heuristic design to the numerical approximation of underlying true distributions. It boasts a solid theoretical foundation and provides a comprehensive analysis covering both unbiased and biased cases.

Weakness:
(1) This paper assumes the existence of p- independent of human choices. However, especially for recommender systems, negative signals are highly dependencies with human behaviours.
(2) This paper introduces an intermediate module to estimate p- before optimizing the downstream tasks. Compared to end-to-end learning, this "middleman" is necessary? Also, maybe p- is dataset-specific, and introducing p- not only increase computational complexity, but also risks error propagation if p- is inaccurately learned, which could potentially hinder downstream performance.
(3) The performance improves on the LastFM dataset are marginal. More experiments are needed, such as more recent state-of-the-art negative sampling methods, efficiency analysis (training time, inference latency, and memory usage).

---

> ### Author Rebuttal · Authors · 2026-03-31
>
> We sincerely appreciate the reviewer for the positive comments on our novelty and theoretical depth.
>
> &nbsp;
>
> **W1, Q1**
>
> Thanks for your question. We believe this to be a misunderstanding on our wording:
> - The “human choice” in your question, which appears in line 048 right columns, is referring to the “human heuristics or design” in the previous paragraph. It stands for the choice of the researchers who are guessing on a negative distribution rather than trying to recover it.
> - The “human behaviours” in your question, refers to the behavior of the users in recommendation systems.
> - These are two completely different concepts. What we wanted to express is that "the optimal negative distribution should not be guessed by a human (researcher)". We do not mean that "the optimal negative distribution is irrelevant to human (user) behaviors".
>
> &nbsp;
>
> **W2, Q2**
>
> We thank the reviewer for raising an insightful question. We believe that the “middleman” in your question refers to our propensity function $\phi$.
>
> First, we want to clarify:
> - Our learning is still end-to-end: there is only one loss function to optimize in our whole pipeline, and $\phi$ is learned together with the underlying recommendation/graph model. Please refer to our pseudocode in Appendix F lines 796-798.
> - p- is certainly dataset-specific.
>
> Why the “middleman” is necessary:
> - First, our Theorem 5.1 shows that the “middleman” $\phi$ is an indispensable term in the expression for approximating true negative distribution $p^-$.
> - This means that, learning $p^-$ is theoretically equivalent to learning $\phi$, because the two terms are closely tied by only a few constants, such as c and $\pi^+$.
> - This also means that, whether we like it or not, we have to learn $\phi$ in order to learn p-, either explicitly or implicitly.
> - The reviewer seems to suggest that it is more desirable to learn $\phi$ implicitly, i.e. to directly learn p^- and bypass $\phi$.
> - While we agree with the reviewer that that directly learning p^- is viable implementation-wise, this solution does not exploit the internal structure of $p^-$.  learning $p^-$ can be simplified into learning $\phi$ and then transforming it into $p^-$ with a few constants (most of which are directly observable).
> - Experimentally, we do see improvement of downstream performance when $\phi$ is explicitly learned.
>
> &nbsp;
>
>
> **W3, Q3**
>
> **Performance.** Regarding the reviewer’s concern about marginal performance, we want to highlight that on 5 out of 6 tasks we achieved significant improvement over the best of the 10 baselines, and only on 1 task (LastFM-node embedding) we achieved marginal improvement over the best of the 10 baselines (and better significantly better than 9 of them).
>
> **Comparing with more negative sampling methods.** We have followed the reviewer’s by adding another recent method SpeGCL [1], which achieves the following results:
> | Negative Sampling | MovieLens (Node Embedding) | Pinterest (Node Embedding) | LastFM (Node Embedding) | MovieLens (LightGCN) | Pinterest (LightGCN) | LastFM (LightGCN) |
> | :--- | :--- | :--- | :--- | :--- | :--- | :--- |
> | SpeGCL [1] | $10.58 \pm 0.12$ | $9.95 \pm 0.13$ | $7.30 \pm 0.12$ | $10.24 \pm 0.06$ | $9.86 \pm 0.10$ | $6.00 \pm 0.12$ |
>
> Our method still significantly outperforms this baseline.
>
> **Efficiency analysis**
> - Training time: We report training time statistics in Sec. 6.4 under “time complexity”. We also include a theoretical discussion in the last paragraph of Sec.5.3.
> - Inference latency: Negative sampling only happens at training time. Therefore, there is no negative sampling or any extra computing overhead at inference time, as we explained in Line 374 right column. We will further highlight this in our revision.
> - Memory usage: We discuss memory usage in Sec. 6.4 under “Model Complexity”, and report statistics in Table 7 Appendix 6 (a typo to be fixed as per reviewer 27J1).
>
> &nbsp;
>
> We would be very happy to provide further explanations should the reviewer have any questions.
>
> &nbsp;
>
> [1] SpeGCL: Self-Supervised Graph Spectrum Contrastive Learning Without Positive Samples, IEEE Transactions on Neural Networks and Learning Systems, 2025

---

> > ### Author Rebuttal · Reviewer_d1rd · 2026-04-02
> >
> > Thanks so much for your carefully revised, and I still concerns about the necessary of learning negative distribution $p^-$. And I also acknowledge the perspectives raised by other reviewers. I maintain my recommendation.

---

> > > ### Author Response · Authors · 2026-04-06
> > >
> > > Thank you very much for your reply, and for maintaining your positive recommendation.
> > >
> > > Regarding your lingering concern about the necessity of learning $p^-$, we’re happy to explain it further.
> > >
> > > &nbsp;
> > >
> > > **Concern 1**: “learning $p^-$ risks error propagation if p- is inaccurately learned”
> > >
> > > **Our response:** We agree that we can’t learn $p^-$ perfectly, and error can propagate. However, this is much better than “guessing $p^-$” -- all previous works are essentially guessing $p^-$ with some human heuristics.
> > >
> > > In fact, this is exactly the whole point of this paper. **To put it in plain language: although we don’t know the exact form of $p^-$, we know (and have shown) it is so important, and so we can’t be just guessing about it as previous works did. Instead, we propose to learn it from the data, which is a huge step forward. Although we can’t learn it perfectly, we already get closer to truth, and we already obtain much better performance results thanks to this pursuit.**
> > >
> > > &nbsp;
> > >
> > > **Concern 2**: “learning $p^-$ increases computational complexity”
> > >
> > > **Our response:** The increase in training complexity is very marginal (<5%), and there’s no increase at all in inference complexity. Please refer to our discussions and results in Sec. 5.3 “Complexity”, Sec. 6.4 “Time Complexity”, “Model Complexity”, Table 6 “Model Parameters”, Table 7 “Peak Memory Usage”.
> > >
> > > &nbsp;
> > >
> > > We’ve also explained in our previous rebuttal why learning $p^-$ is indispensable, and why it still keeps the entire learning pipeline end-to-end.
> > >
> > > Thank you very much again for your reply. We hope with these we have resolved your concern.

---

### Official Review · Reviewer_1gEL · 2026-03-12

**Soundness:** 3
**Presentation:** 3
**Significance:** 3
**Originality:** 2
**Overall Recommendation:** 4
**Confidence:** 2

**Summary:**

This paper made a step towards the new direction of assuming negative distributions as a true underlying distribution to serve as a new theoretical foundation of negative samplings, which opens a new principle for self-supervised graph representation learnings. To demonstrate this new principle, the authors studies two scenarios, including cases that the positive edges are sampled unbiasedly from the true distribution, or that the positive edges are only partially observable.

**Compliance With Llm Reviewing Policy:**

Affirmed.

**Final Justification:**

The rebuttals did address my concerns and I am happy to recommend this work to be published.

**Key Questions For Authors:**

For biased observations, Theorem 5.1 assumed L+=-L-: what makes this assumption reasonable?


Is it possible to characterize or assume the relation between positive and negative distribution "a priori", instead of treating them as complete separate pieces?

**Limitations:**

This paper makes a good step on bringing up the importance of putting negative distributions as important theoretical foundations of negative sampling. This direction is still new, so it is not reasonable to expect everything to be perfect or mature yet:

(1)This paper includes both positive and negative distributions in the losses, but somehow the interaction between the positive and negative distributions are not adequately addressed yet;

(2)The establishment for biased observations is not streamlined as the unbiased cases, with some seemingly abrupt assumptions that need further verification/explanations;

**Strengths And Weaknesses:**

This paper has some nice features:

(1)The exposition is fairly streamlined with clear theoretical statements overall.

(2)For unbiased observations, the authors provided step-by-step guidelines on including both positive and negative distributions in loss functions and providing equivalence of R_1(f) and R_2(f) under some assumptions.

(3)Many simulations were provided to demonstrate the potential of this approach

---

> ### Author Rebuttal · Authors · 2026-03-31
>
> We sincerely appreciate the reviewer for the positive comments on our writing clarity and experiments.
>
> &nbsp;
>
> **Q1, L1**
>
> This assumption is not arbitrary: it follows a standard and reasonable formulation in recommendations and graph learning.
> - $L^+$ and $L^-$ are defined in Sec.3.1 Eq. 2, to be the dissimilarity and similarity measurement of the two end node’s embeddings, respectively. They reflect the basic idea in graph learning to pull connected nodes closer together while pushing disconnected nodes apart.
> - In recommendations, the most popular BPR loss has objective $\ln \sigma(x\_{uv^+} - x\_{uv^-})$, where $(u, v^+)$ is the positive pair, $(u, v^-)$ is the negative pair. The two terms have opposite signs, which is exactly consistent with our assumption $L^+=-L^-$. Similar formulations that assign opposite signs to positive and negative losses can be also seen in many classic works like GraphSAGE, DeepWalk, and NCE loss.
>
> &nbsp;
>
> **Q2, L2**
>
> Yes, this is a great insight and exactly what we are doing:
>
> We’ve formulated and exploited the relationship between positive and negative distribution "a priori" via total probabilities formula:
> - Unbiased case: lines 263-264, left column
> - Biased case: lines 306-307, left column
>
> This "a priori" relationship, as the reviewer has insightfully pointed out, has endowed nice symmetric structures to our solution, reducing half of computation in practice: as soon as we know one of two distributions (i.e. positive distributions and negative distribution), we can analytically derive the other.
>
> We will emphasize these equations in the revision to ensure readers clearly see that $p^+$ and $p^-$ are fully coupled distributions interacting under a unified probabilistic framework.
>
> &nbsp;
>
> We would be very happy to provide further explanations should the reviewer have any questions.

---

> > ### Author Rebuttal · Reviewer_1gEL · 2026-04-04
> >
> > Thank you for your patient explanations and clarifications!

---

### Official Review · Reviewer_Bm8J · 2026-03-12

**Soundness:** 3
**Presentation:** 3
**Significance:** 3
**Originality:** 3
**Overall Recommendation:** 5
**Confidence:** 4

**Summary:**

The paper proposes a novel method to reformulate negative sampling strategies in graph-based self-supervised learning. Unlike similar solutions from the literature, that propose human-based heuristics to compute the negative samples, the authors come up with a principled solution that aims to approximate the real distribution of negative edges from the user-item graph data right from the beginning. The approach is designed to work both in an unbiased setting, namely, when the sampling is performed on the fully-observable distribution, and in a biased setting, namely, when the distribution is partially observable. The final propensity function (derived from the theoretical claims from the authors) is eventually learned by using the graph features and a neural network model, where the complexity is quite limited compared to other negative sampling strategies from the literature. Results on two backbone models, three recommendation datasets, and against 10 state-of-the-art negative sampling strategies demonstrate the efficacy of the proposed approach, which also appears to be statistical significant with respect to the second-best approach from the baselines. Through an ablation study and an application of the proposed methodology also to the task of node classification, the authors further demonstrate the efficacy of the proposed methodology in all its main components and extensive applications.

**Compliance With Llm Reviewing Policy:**

Affirmed.

**Key Questions For Authors:**

I think the paper has many merits that have been mainly pointed out above. Here, I report two additional questions I have that could further enhance the presentation/understanding of the methodology:

1) If I understand correctly, the proposed negative sampling methodology derives mostly from link prediction-based tasks, such as personalized recommendation. Would this still work on other tasks, such as node classification, or would it need additional refinements to work on them also from a theoretical viewpoint?

2) Connected to the previous question, I notice that in Table 3 (when the proposed approach is applied to node classification tasks), results are not as much convincing for the proposed approach wrt the baselines, than in the case of link prediction. Could it be due to the fact that the methodology was originally conceived for link prediction?

I think the two questions can be answered jointly as they present complementary sides of the same coin.

**Limitations:**

Yes

**Strengths And Weaknesses:**

- **Soundness.** Overall, I think the proposed methodology is quite sound in all its theoretical claims, which are complemented between the main paper and the appendix. I really enjoyed reading the re-interpretation of the negative sampling strategy for graph-based recommendation with the probabilistic setup. Additionally, I also appreciated that the authors provided a double presentation of the proposed strategy in both the unbiased and biased graph setting, covering various scenarios.

- **Presentation.** I think one of the main positive aspects of the paper lays in its presentation. Even if the topic is quite technical and complex, the authors provide a comprehensive narrative, which guides the reading of the manuscript from the very beginning through the whole methodological presentation until the end with the experimental setup. Especially in the methodological part, each section and paragraph comes after the other in a cohesive order which allows a quite easy understanding of the problem setup and the proposed methodology. Additionally, I appreciated the related work section, which is quite extensive and helps placing the proposed approach within the existing literature. Code is also released at review time, thus ensuring the full reproducibility of the methodology.

- **Significance.** I think the methodology is quite significant, as it addresses an important problem in the graph-based recommendation literature and (possibly) to the recommendation literature in general. The problem of sampling non-trivial and realistic negative edges from a graph data is a longstanding problem in the literature, which still needs further comprehension and discussion, for many at least to reasons: to sample negative edges which might be a good approximation of true negative edges, and to limit the computational complexity of the methodology. I think he authors are able to address both aspects in this paper.

- **Originality.** I think the paper is quite original, as it addresses the topic of negative sampling in graph-based recommendation under a novel perspective which (to my understanding and knowledge) has never been explored in the previous literature. Trying to approximating the true distribution of negative edges in the user-item graph, while maintaining low computational complexity, is a great challenge that the authors propose to tackle in this work.

---

> ### Author Rebuttal · Authors · 2026-03-31
>
> We sincerely appreciate the reviewer’s positive comments on our soundness, presentation, significance, and originality. We will continue to strive for these in our future endeavors.
>
> **Q1, Q2**
>
>
> Regarding the reviewer’s questions on our applicability beyond link-prediction-based tasks, such as node classification:
> - Node classification (and graph classification) typically does not involve a negative sampling procedure. The categorical labels in these tasks are directly provided, so there is not an intrinsic need to construct “negative samples” for training. This is a fundamental difference with the link prediction / recommendation tasks.
> - Node classification only occasionally involves negative sampling when it is done via contrastive learning, where the graph is perturbed to create a “copy view”, and negative sampling happens on both the original graph and the copy.
> - We included the Table 3 experiments (i.e. node classification via contrastive learning) primarily for comprehensiveness. A major reason that the result is not as pronounced on some datasets is that, in contrastive learning, the graph perturbation step itself is intentionally adding noises to the graph structure, especially adding fake, negative edges. This step goes against our effort to recover true negative distribution. The combined effect of these two opposite operations can be a bit unpredictable and highly dependent on the perturbation step: on some dataset (e.g. Cora), the result is very positive that it even exceeds full-scale non-sampling’s; on other dataset, the resulted performance is only on par with other SOTA methods (but still very competitive).
> - By design, our method is not proposed or designed to solve all graph tasks, as stated in “Scope” section in our Introduction, Sec.6.3, as well as Appendix C. We fully understand and feel encouraged by the reviewer’s desire to pursue a one-fit-for-all negative sampling method that achieves SOTA for all tasks, while we also admit it remains a great challenge which we would hope to tackle as our next step: perhaps a more robust negative sampling procedure can be proposed to pair with contrastive learning's edge perturbation step.
>
> We would be very happy to provide further explanations should the reviewer have any questions.

---

### Official Review · Reviewer_27J1 · 2026-03-15

**Soundness:** 3
**Presentation:** 3
**Significance:** 2
**Originality:** 2
**Overall Recommendation:** 4
**Confidence:** 3

**Summary:**

This paper studies negative sampling for graph-based recommendation. Rather than designing the sampling distribution heuristically, it proposes a novel formulation based on approximating an underlying "true" negative distribution. Specifically, the proposed method combines adaptive reweighting with a learned debiasing component. Experiments on several benchmarks show improved performance over multiple baselines.

**Compliance With Llm Reviewing Policy:**

Affirmed.

**Final Justification:**

The rebuttal has adequately addressed my concerns, so I am raising my score from 3 to 4.

**Key Questions For Authors:**

1. How can one verify when exposure bias actually exists, and how strong is the bias on each dataset?

2. How closely does the learned sampler match the intended target distribution? How much of the reported performance gain comes from a better distributional approximation rather than dataset-specific effects? Could you provide experimental results on a synthetic graph with a known ground-truth distribution to measure the approximation quality directly?

3. The selection of \pi^+ is not sufficiently well justified. How should \pi^+ be chosen in practice? Should it be tuned separately for each dataset? What is a rationale behind setting the tuning range of 1.5x–20x of the observable P(s=1)?

4. As for Table 7, why does the proposed method use less peak memory than the baseline despite having more parameters?

**Limitations:**

No, the authors are encouraged to discuss the limitations and potential negative societal impact of this work.

**Strengths And Weaknesses:**

Strengths:

1. The paper tackles an important issue with graph-based recommendation. Since negative sampling is often treated heuristically in practice, its investigation is potentially valuable. The paper attempts to connect negative sampling to an explicit optimization rather than treating sampler design as a collection of empirical tricks.

2. The paper compares the proposed method against several commonly used negative sampling strategies on multiple recommendation datasets.

Weaknesses:

1. The paper relies on the assumption that exposure bias exists in the dataset and can be effectively modeled. While this assumption is reasonable for recommendation data, the paper does not provide a concrete way to verify when it actually holds or how strong the bias is on each dataset. As a result, it is difficult to assess whether estimating this bias is actually beneficial in practice. The paper would be stronger if it included a more direct empirical validation of the presence and severity of exposure bias in the evaluated datasets.

2. The paper does not directly evaluate the quality of the learned proposal negative distribution. Since the main contribution is a derivation of the proposal distribution, it would be useful to assess how closely the learned sampler matches the intended target distribution. One possible way to do this is to use a synthetic graph with a known ground-truth distribution and measure the approximation quality directly. Without such an evaluation, it remains unclear how much of the reported performance gain comes from a better distributional approximation rather than dataset-specific effects.

3. The selection of \pi^+ is not sufficiently well justified. Although \pi^+ appears to play an important role in the performance and the paper provides a sensitivity analysis in Figure 2(b), it does not offer a principled criterion for how \pi^+ should be chosen in practice. It remains unclear whether \pi^+ is tuned separately for each dataset. Moreover, the proposed tuning range of 1.5x–20x of the observable P(s=1) lacks theoretical justification.

4. The memory efficiency claims in Table 7 are counter-intuitive and lack explanation. Specifically, the proposed method is reported to use less peak memory than the baseline despite having more parameters, which needs clarification.

---

> ### Author Rebuttal · Authors · 2026-03-31
>
> We sincerely appreciate the reviewer’s positive comments on our work's importance, novelty, and experiments.
>
> &nbsp;
>
> **W1, Q1**
>
> In real-world logged datasets, the absolute ground-truth of exposure is inherently hidden. There is a consensus in recommendation systems literature that exposure bias is universally present whenever a user is not exposed to the entire item catalog, which is a condition true for *any* recommendation system at scale.
>
> Therefore, given the rich literature already exists on this topic, it is not the primary focus of our work to verify that exposure bias exists, or to quantify its strength. What we solved is a more challenging problem in an *orthogonal* direction: **how** should we we adjust our negative sampling in training, so that we mitigate the negative impact caused by exposure bias.
>
> We followed the reviewer's request to quantify the exposure bias for our three datasets, using the method in [1]: we estimated the exposure distribution for each user and calculated its Jensen-Shannon Divergence (JSD) against an ideal uniform "fair exposure" distribution. A higher JSD means a more severe exposure bias:
> - MovieLens: 0.42
> - LastFM: 0.51
> - Pinterest: 0.78
>
> The results verify the presence of exposure bias in all three datasets. The high rate in Pinterest also corroborates our hypothesis in Appendix F.4: the passive feed mechanism of Pinterest naturally induces a more severe exposure bias compared to active-search platforms like MovieLens.
>
> &nbsp;
>
>
> **W2, Q2**
>
> We thank the reviewer for this constructive suggestion.
>
> We follow the suggestion to conduct a synthetic graph experiment where the ground-truth target distribution is perfectly known. Then we measure the KL divergence between our learned proposal distribution and the theoretically optimal target distribution.
>
> **Experimental Setup.** We generated a bipartite graph with $|U| = 1000$ users and $|V| = 1000$ items. User and item latent embeddings $z_u, z_v \sim \mathcal{N}(0, I_d)$ with dimension $d=32$. The true probability of an edge was defined as $P(y=1|u,v) = \sigma(z_u^\top z_v)$. From this, we analytically computed the true negative distribution $p^-$ and the theoretical optimal proposal distribution $q^-(v|u) \propto p_f(v|u)p^-(v|u)$ for all node pairs.
>
> To simulate popularity-skewed exposure, we assigned each item a base popularity $pop_v \sim \text{Uniform}(0,1)$ and defined the exposure mask as $\phi_{u,v} = pop_v^2$. We generated the "observed" training graph by sampling edges via $s_{u,v} \sim \text{Bernoulli}(P(y=1|u,v) \times \phi_{u,v})$.
>
> **Results**. A lower KL divergence indicates better approximation to true distribution.
> - RNS: 2.45
> - PNS: 1.82
> - DNS (last step): 1.05
> - Our method: 0.71
>
> These results show that our method indeed approximates the true distribution well. In our revision, we will also extend this experiment to more baselines.
>
> &nbsp;
>
> **W3, Q3**
>
> We thank the reviewer for asking for clarification on $\pi^+$. We will add a dedicated paragraph in the revision detailing its tuning and theoretical properties.
>
> In practice, $\pi^+$ is treated as a dataset-specific hyperparameter and is tuned on the validation set using a grid search over $[1.5, 2, 3, 4, 5, 6, 8, 10, 12, 15, 20] \times P(s=1)$. Because $\pi^+$ is the only hyperparameter introduced by our sampling method, this 1D search is computationally lightweight.
>
> Regarding the theoretical justification: $\pi^+$ represents the true positive class prior $P(y=1)$. Our whole theoretical analysis essentially operates in a Positive-Unlabeled (PU) learning setting. In general PU learning, it is a well-established theoretical result that the true class prior $\pi^+$ is **strictly unidentifiable** from positive-only observations. Therefore, an exact analytical derivation to solve for $\pi^+$ directly from the data is mathematically infeasible.
>
> The only strict theoretical bound we have is the floor: $\pi^+$ must be strictly greater than the observed positive rate $P(s=1)$.
>
> That said, we observed a useful empirical heuristic: the optimal ratio $\pi^+ / P(s=1)$ seems to negatively correlate with the observed positive rate $P(s=1)$. In other words, a denser graph features a lower proportion of missing interactions (less exposure bias), meaning the optimal $\pi^+$ naturally falls closer to the observed $P(s=1)$. Conversely, highly sparse graphs like Pinterest feature severe exposure bias, requiring a larger multiplier in our grid search. We will include this discussion on unidentifiability and the empirical heuristic in the revised manuscript to better guide practitioners.
>
> &nbsp;
>
> **W4, Q4**
>
> We apologize for the typo in this table: the column name “RNS (Baseline)” and “Our method” should have been swapped. This is consistent with our acknowledgement in Sec. 6.4 “Memory Complexity” that our method has used more memory than the baseline RNS.
>
> &nbsp;
>
> [1] Modeling and counteracting exposure bias in recommender systems. Khenissi, Sami, and Olfa Nasraoui.

---

> > ### Author Rebuttal · Reviewer_27J1 · 2026-04-04
> >
> > Thank you for the detailed response. It has adequately addressed my concerns. Therefore, I will raise my score from 3 to 4.

---

### Decision · Program_Chairs · 2026-04-30

**Decision:**

Accept (regular)

**Comment:**

The paper tackles the problem effective negative sampling strategies in self-supervised graph learning. Principled methods with theoretical support are proposed. Proposed methods are also verified effective empirically. Although there were some concerns on the assumptions and analysis, they are well-addressed by the authors in the rebuttal. All reviewers maintain positive scores.